# Proactive Corporate Sustainability via Social Innovation— A Case Study of the Hennes & Mauritz Grand Challenge in Bangladesh

**Veronika Tarnovskaya** [1], **Sara Melén Hånell** [2,*] and **Daniel Tolstoy** [2]

1 Department of Business Administration, Lund School of Economics and Management, Lund University, 223 63 Lund, Sweden; veronika.tarnovskaya@fek.lu.se

2 Department of Marketing and Strategy, Stockholm School of Economics, 113 83 Stockholm, Sweden; daniel.tolstoy@hhs.se

\* Correspondence: sara.melen@hhs.se

**Abstract:** The purpose of the study is to explore how a multinational enterprise can use social innovations to drive change and solve grand challenges in an emerging market context. This paper brings market-shaping literature into a sustainability context, particularly by studying the implementation of social innovations in an emerging market context. Specifically, the study involves an in-depth qualitative study of H&M's fair living wages program in Bangladesh. We find that H&M is tackling utterances of grand challenges revealed by orchestrating social innovation in collaborations with local stakeholders. Social innovation is carried out in ongoing projects involving multiple stakeholders. The study contributes to current literature by revealing that multinational enterprises indeed can use social innovation to drive change in emerging markets, although this requires long-term commitment, an ability and willingness to shape the surrounding business environment, and a prominent standing among key stakeholders.

**Keywords:** corporate sustainability; market shaping; social innovation; proactive CS; emerging markets

## 1. Introduction

Multinational enterprises (MNEs) are to an increasing extent expected to behave in accordance with ethical principles and have, in recent decades, been held accountable for their actions by customers, shareholders, nongovernmental organizations, and lawmakers [1–4]. One such example is the Swedish fast-fashion company Hennes & Mauritz (H&M), which has been continually scrutinized and, at times, exposed to public outrage for environmental harm and labour exploitation in its production markets in Southeast Asia. Supply chain management is a critical competitive factor for H&M to maintain market responsiveness and cost advantages. Complying with sustainability guidelines in supply chain operations has become a strategic priority for the company to avoid reputational damage but also to develop the brand. Research has pointed at an increasing number of MNEs that, like H&M, are treating corporate sustainability (CS) as means to gain a competitive edge [5–8]. In this article, we particularly focus on H&M, and we explore how the company implements CS in its operations in Bangladesh, being the company's largest production market.

CS is, in this article, understood as the organizational behaviour aimed at balancing social, economic, and environmental goals [9]. CS involves both a reactive behaviour of handling risks of noncompliance to sustainability standards, as well as proactive behaviour, which could comprise efforts to push the sustainability agenda to create value outside the realm of the company's immediate business operations [10]. The proactive approach to sustainability has become increasingly relevant, along with managers placing sustainability operations at the heart of their business strategies [11] and, largely, treating it as a value

proposition to MNE's internal and external stakeholders. In this way, MNEs are actively using their sustainability-driven value propositions as market-shaping devices allowing them to mould and manage the evolution of markets and initiate change of structure of markets and behaviours of market actors [12,13].

The market-shaping phenomenon has been extensively explored in the marketing and international business (IB) literature as a market-driving strategy [14], a proactive market orientation [15], market innovation [16], and market-shaping [17]. One of the major postulates of the market-shaping approach is that it brings about a set of activities that may reshape the value chain and change behaviours of key actors, such as customers, competitors, suppliers, and other local stakeholders [14]. Despite such a focus, the link between market-shaping and sustainability operations has surprisingly neither been sufficiently acknowledged or empirically substantiated in international business settings. In this article, we bring market-shaping literature into a sustainability context, particularly by studying the implementation of social innovations in an emerging market (EM) context.

The role of innovation is emphasized in the market-shaping literature as the driving force behind the leap in value proposition to customers [18]. It is also suggested that a radical business innovation needs to be put in place. On the other hand, Storbacka and Nenonen [19] proposed a softer and more subtle approach, in which focal firms offer "market propositions" that engage other actors in creating a shared market view, which, in turn, leads to changes in the mental models of all actors in the market. The IB literature proposes that social innovations can function as such propositions, as they benefit a broad group of actors, emerge through interactions between firm and market actors, and create necessary conditions for the positive life changes for many people [10]. While MNEs may have the resources and legitimacy to drive social innovation programmes that have an impact on developing societies, this issue remains largely unexplored. In comparison to corporate social strategies that outline a framework for the minimum level of compliance, social innovation programmes go beyond that and may lead to more progressive outcomes. We think this topic deserves more attention, since combining public and private efforts to create social change may be a way to solve the grand challenges of the 21st century.

In this article, we focus on MNE operations in Ems, as these are the most relevant contexts for our study due to the gravity of ethical and environmental problems in these settings, sometimes considered grand challenges. The purpose of this study is to explore how a MNE can use social innovations to drive change and solve grand challenges in an EM context. By "change", we mean the behavioural changes of key actors in the value chain and structural changes in the industry caused by MNE operations in EM. Aligned with the purpose of the study, we conducted a case study on H&M and its work on implementing fair living wages in the EM of Bangladesh, the largest production market for the company. The case study centres on illustrating the specific challenges that the MNE has encountered in the implementation process and those activities that it has undertaken to manage and overcome those challenges.

This study attempts to contribute theoretically, as well as empirically, to the IB literature. First, we contribute to IB literature by theorizing on the role of social innovations as important means to drive change and shape markets that are characterized by volatility, unstable laws, often-inferior standards and weak market institutions, and when the wellbeing of diverse local stakeholders is crucial for a MNE's CS agenda. Empirically, we explore and illustrate the role of social innovation used by MNEs for tackling both minor sustainability issues, as well as grand challenges [10,20] that often cannot be solved by one focal firm alone and need collaboration with other firms and local actors via a proactive approach. We focus on the specific type of social innovation—mission-oriented innovation that is usually correlated with the focal firm's mission and value proposition. In sum, by theorizing and empirically illustrating social innovation as the means for proactive CS by MNE in the EM setting, we can broaden the IB literature with a phenomena-driven and broader interdisciplinary perspective as regards grand challenges facing both MNEs and societies.

## 2. Theoretical Framework

### 2.1. Proactive CS as a Response to Grand Challenges

In their call for the renaissance of IB research, Buckley et al. [10] addressed one of the most important emerging phenomena in the field—the increasing pressures for social responsibility and sustainability that MNEs are facing in their global operations. One of the ways to address this "grand challenge" at the corporate level is to take a proactive approach by developing the sustainability agenda to an extent that reaches outside the company's immediate business operations. Grand challenges are defined as "ambitious but achievable objectives that harness science, technology, and innovation to solve important national or global problems and that have the potential to capture the public's imagination" [21]. Among these challenges are broader issues such as climate change, poverty, migration, and health. "Decent work to all" (including fair wages) is, for example, one of the grand challenges that is especially relevant for the empirical study in this article (UN Sustainable Development Goals 2030, Goal 8).

Grand challenges are examples of problems that transcend geographical and economic borders and affect both MNEs and societies in which they are embedded. Tackling these challenges requires widening the scope of the CS agenda. When pursuing grand challenges, CS cannot merely be a defensive response to external pressures, risk mitigation, and damage control. Arguably, companies need to shift mindsets to a decidedly strategic focus and proactive approach to come to terms with structures underlying social and environmental problems.

The proactive CS (of CSR) approach is concerned with MNEs' focus and responsibility for broader societal issues and different ways of managing stakeholder relationships. The importance of managing stakeholder relationships on a broad scale has been illustrated in several studies. In these studies, stakeholder pressure is described as the driving force of a firm's CS and CSR pursuits. Among these pressures are influences of customers [22], employees [5], investors [23], or society [24]. Thus, firms pursue a CS agenda only after they have been exposed or pressured by their external stakeholders and society. From this perspective, CS activities are predominately reactive. However, firms also engage in CSR for the reasons of philanthropy, to increase transparency, enhance firm reputation [25], to improve trust and cooperation [26], develop stronger ties with local governments [27] and, in doing so, enhance the firm value by taking a leadership position.

In EM settings, MNEs have been seen to take on such proactive approaches in driving change through initiatives that affect the wellbeing of local communities and the work conditions in local industries [28]. Although there is increasing awareness on the embeddedness of MNEs within their local, regional, and global context [29] and the nature of interactions among MNEs and external actors, studies unpacking the MNE's proactive responses to local and global challenges are few. For example, Regner and Edman [30] were able to identify the factors enabling MNE subunits to shape the local institutional contexts to their advantage, but the study still focused on the focal firm and did not take into account the broader societal perspective.

Consequently, a deeper understanding of proactive CS as a market-shaping approach might be a way to understand MNEs' responsiveness to societal problems ingrained in local communities, in turn affecting economic life. As argued in Buckley et al. [10], it is vital that IB research establishes a stronger connection to the contextual characteristics of MNE's environment to not only address the economic performance but also the performance in relation to grand challenges. The empirical case study conducted in this study, featuring a specific grand challenge related to fair wages, adheres to such concerns.

### 2.2. Market-Shaping Research

Market shaping is increasingly being recognized as a viable and deliberate market strategy [31,32] that treats markets as structures that are being continuously shaped and reshaped by actors. This view stands in contrast to the dominant view of markets to be "out there" as static entities to be acted upon. Market shaping aims at influencing markets by means of diverse activities, covering a wider scope of actors than just customers and

competitors in pursuit of a sustainable competitive advantage [33,34]. Researchers in the field have introduced various concepts to describe proactive market shaping. Among these concepts are market driving [14], market scripting [35], market innovation [36], and market shaping [17]. All these concepts describe processes of the formation of markets and specific changes to the market structure and behaviours of market actors.

The specific focus on activities as vehicles for shaping markets is acknowledged in the literature [14,32]. Market-shaping activities cover a broad range of exchange-related, institutional, and technological activities deployed by the main actor (e.g., MNE) to influence a target market. They can be pursued on different levels of influence: the system level, the market offer level, and the technology level [32]. The system level is concerned with the norms and regulations that set the boundaries and rules for the entire market, which is not reduced to customers and/or competitors but also includes downstream actors [32], implying a focus on institutions and institutional arrangements [37]. The market offer level is about the exchange object, the value proposition, or market offer, including products and services that make that offer. Market shaping (driving firm) is argued to create a discontinuous leap in value proposition to a broad group of stakeholders, allowing it to outperform its rivals [14,18]. A market-shaping strategy involves close interactions between market actors, within which a value proposition emerges. In a study on global supplier relationships, Elg et al. [31] found that a successful market-shaping strategy depends on supplier's resource sharing with other actors in the channel, thus enhancing the value proposition.

The technology level of market shaping indicates the role of technology as a functional base for the shaping of different activities and creation of market offers. It usually consists of new technology or business innovations, but it can also include social innovations. Indeed, Vargo et al. [38] argued that "technology both physical and social can be conceptualized as potentially useful knowledge that may provide solutions for new and existing problems" (p. 65). Surprisingly, the role of social innovations as vehicles for market shaping has not been discussed in the marketing literature while given more attention in the IB literature [39]. In this study, we used the activities framework from the literature to uncover the specific activities carried out by MNC in the EM, with a special focus on the social innovations enabling a firm to create lasting changes in market actors' behaviours, as well as structural changes in the industry.

*2.3. Social Innovation for Market Shaping in EMs*

The concept of social innovation has entered both policy debates and is also gaining traction in academic research [28,40,41]. Kanter [42] described social innovation as springing from a mindset that regards community needs as opportunities to develop ideas to find and serve new markets and to solve long-standing business problems. Existing research about social innovation remains fragmented [43]. Chin et al. [41] emphasized that social innovation is a multifaceted concept that involves a wide range of activities that may differ in nature, scope, and targeted objectives. Several scholars have identified the main difference between social and business innovation to be the intended outcome [43]. While economic value creation is a key target of business innovation, social innovations are directed towards values that benefit society as a whole [28]. The outcomes of social innovations might be both material (products, services, and technology) and immaterial (processes, modes of organization, and markets). Moreover, outcomes of social innovation may simultaneously meet a social need and lead to new or improved technological capabilities and/or relationships [44] (p. 18). In this way, social innovations can both benefit society and the focal firm.

Recently, the literature has drawn a line between social innovation and efforts to proactively tackle grand challenges, for example, achieving UN's social development goals [45]. Prashantham and Birkinshaw [46] also argued that MNEs can benefit societies—and themselves—by cooperating with local companies in Ems, where addressing SDGs is a pressing concern. Developing local relationships for this purpose would allow firms to combine complementary resources to address societal and economic needs simultaneously.

Correspondingly, Nylund et al. [45] argued that, to develop social innovations, MNEs form business ecosystems and networks involving local stakeholders that influence not only businesses but wider social systems. However, the literature has not yet fully accounted for this diversity of social innovation, specifically related to the different types of social innovations, as well as how social innovations can serve (or not serve) proactive efforts of CS. To address this issue, we propose that social innovation takes place on a spectrum either homing in on situation-specific or highly localized social problems at one end or focusing on large social problems permeating entire societies at the opposite end. We refer to the social innovations that cover a delimited scope as task-oriented innovations, and we refer to social innovations that cover a relatively wider scope as mission-oriented social innovations. On the one hand, task-oriented social innovations have operational foci and aim to reduce damage, mitigating risks of misconduct and correcting deviances [47]. As such, they are likely to treat the symptoms of social or environmental misconduct in a specific setting. On the other hand, mission-oriented social innovations are directed at bold, inspirational, and ambitious goals that have broad relevance for both business and society [48]. They are delivered through the cocreation of multiple top-down and bottom-up solutions and rely on cross-sectoral activities [49]. According to Mazzucato [48], the most interesting applications around mission-oriented innovation are being driven by the needs of emerging economies.

Furthermore, the role of social innovation in market shaping in general and in proactive CS as a strategic activity has not been given adequate attention in the literature. By focussing on social innovation as a means for proactive CS by MNEs in EM settings, we are answering multiple calls for phenomena-driven and broader interdisciplinary perspectives that suggest MNEs have a role to play in tackling grand challenges affecting societies at large. We propose, in turn, that a proactive market approach may involve moving beyond a risk-mitigating focus related to CS issues and involve an inclination to tackle grand challenges by shaping markets to reap long-term benefits and sustainable change.

In Figure 1, we combine the key tenets of market-shaping research and social innovation for a preliminary framework for the study.

The conceptual framework illustrates how social innovation (mission-oriented and task-oriented) can be used for proactive CS in an EM context, conceptualized as marketing shaping activities, and focus on grand challenge objective(s). This framework is tentative in a sense that it helps us understand how social innovation manifests itself, how it is used by MNC to plan and implement different change-oriented activities aimed at local challenges, and what the outcomes are. It does not imply the direct relationship between social innovation and proactive CS in a way that one leads to the other. We are suggesting that social innovation is an integral part of many MNC CS strategies, but its role is not fully recognized. By specifically aiming at unpacking social innovation, we expect to contribute to the CS literature in the field of international business by examining the role of social innovation as a driver of proactive CS strategies of MNEs.

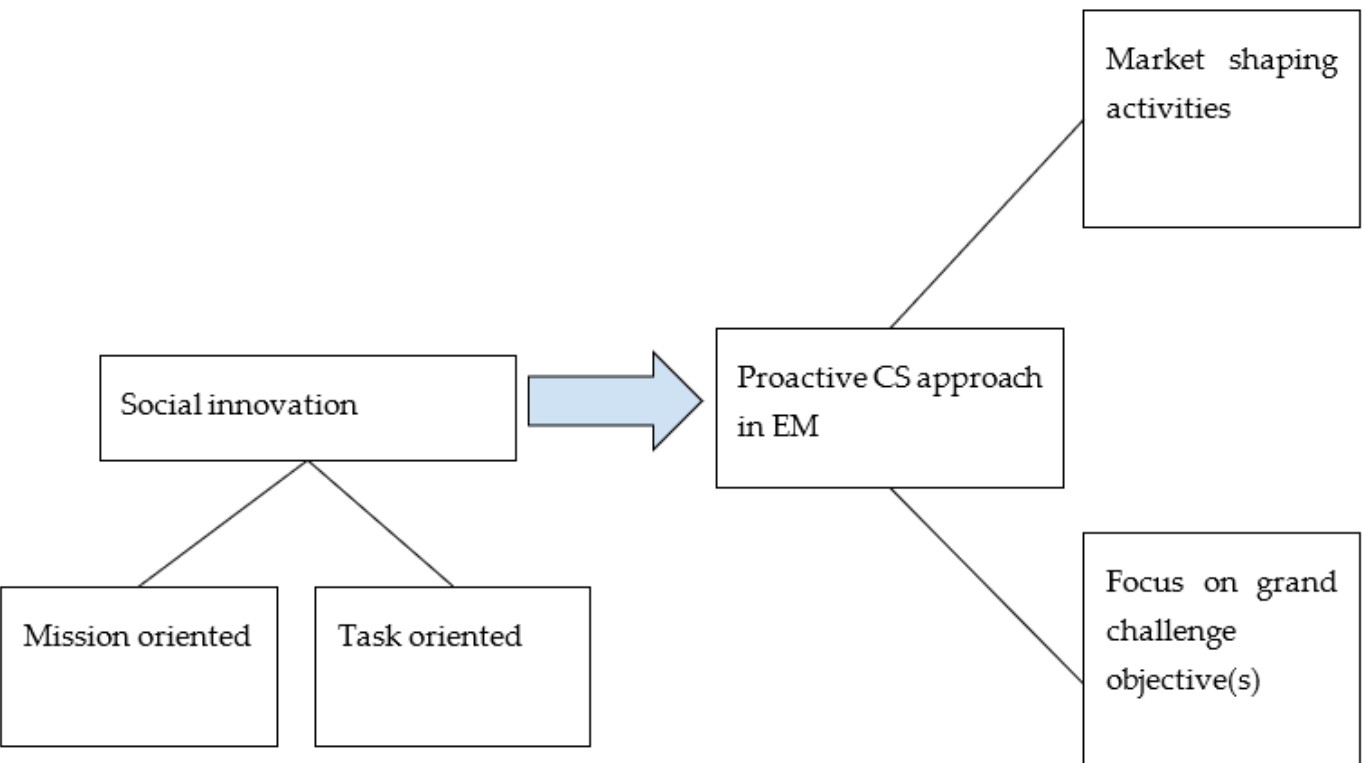

**Figure 1.** Conceptual framework of how social innovation can be a means of proactive CS in an EM context.

### 3. Method

#### 3.1. The In-Depth Qualitative Case Study Approach

This study rests on the premise that there is a need for more empirical, as well as theoretical, research on how MNEs can use social innovations for tackling minor sustainability issues, as well as grand challenges. In specific, calls have been made for phenomena-driven and broader interdisciplinary perspectives regarding grand challenges facing both MNEs and societies. Against this backdrop, we find an exploratory research approach, based on an in-depth qualitative case study approach, most relevant [50,51]. We explored new insights about practices, conceptions, and occurrences, providing a rich understanding of a complex research topic. The empirical data presented in this paper were therefore designed to give an in-depth description of how the global fashion company H&M implements CS in its operations in Bangladesh. The case centred on how this MNE has implemented their work on fair living wages in the EM of Bangladesh. The experiences, narratives, and events related to this process provided a basis for analysis that allowed us to advance the theoretical understanding on the topic

We have, in this paper, followed a purposeful, theoretical sampling strategy [52] to explore the management-related challenges of an MNE's implementation of sustainability practices in an EM context. The global fashion company H&M and its work on implementing fair living wages in Bangladesh was chosen as a focus of the in-depth case for various reasons. H&M is an example of an MNE that has documented experience of working on sustainability matters for a relatively long time-period in emerging markets. One of its largest production markets is in Bangladesh. Bangladesh is also a production market where H&M has been present for a relatively long time. Some of the suppliers in the market have collaborated with H&M for more than 20 years. To gain in-depth insights on those management-related challenges experienced by an MNE in implementing sustainability practices in an EM context, H&M and its operations in Bangladesh thus appeared as an interesting case to study. While H&M CS operations span a wide range of areas, the case

study is focused upon the company's social sustainability strategy and, specifically, their work on introducing "fair jobs" and fair living wages in their production supply chain. We decided to focus on this area due to the rich data that we could gain regarding the specific challenges encountered in the implementation process and those activities that have been undertaken to manage and overcome those challenges. We recognize that this single case study of a firm in the fashion industry restricts us from generalizing the results to other companies. For example, being a global consumer-oriented firm has certain implications in the public scrutiny of CS operations, brand recognition, and social judgments in comparison with, for example, a business-to-business firm. Nonetheless, the quality and richness of data allowed us to capture a pertinent and emerging phenomenon and develop conceptualizations that could serve useful for future studies that seek to develop knowledge and theories about social innovation as a driver of CS development. While consumer-oriented firms currently seem to take the lead in this area, this is a growing phenomenon that, over time, could be applicable on a wider group of firms.

The overall research design and the empirical data collection can be described and illustrated by two phases. In May 2019, we entered the first phase of our research design when we started to conduct interviews with managers at the HQ level at H&M. The aim of this first phase was to investigate and understand the policies and codes of conduct relating to sustainability matters and to study the routines for interacting with local offices in EMs. In the second empirical phase, which began in 2020, we focused particularly on H&M's organization and operations in Bangladesh and how sustainability issues unfold, develop, and how the sustainability work is implemented in this local market.

*3.2. Data Collection and Analysis*

This study was based on both primary and secondary data. The main data source was ten interviews made with managers at the headquarter (HQ) level in Stockholm and the production office located in Bangladesh (see Table 1). Interviews were conducted with those managers at the HQ level and the EM level (i.e., Bangladesh) that were involved with tasks and issues related to the MNE's sustainability work. Five interviews were made with four managers located at the HQ in Stockholm. The interview questions that guided all our interviews at the HQ level concerned the organizational structure of sustainability of the whole MNE. We focused the questions on how sustainability activities are organized and implemented at the HQ level, as well as on the local EM. The questions also covered how the MNE interacts and collaborates with different stakeholders and how different ethical codes and commitments are developed and implemented throughout the organizations and markets.

Five interviews were done with managers in the sustainability team operating in Bangladesh. These interviews were digital, and in each interview occasion, we met up with a team of three or four managers helping us to gain an understanding on the chosen topic. In the first interview with the sustainability team in Bangladesh (June 2020), we focused upon questions on the local sustainability organization and what sustainability issues the local teams were currently working on. In the second interview (August 2020), we focused on the environmental team, with questions about those sustainability issues being the focal point of this local team. The third interview (September 2020) focused on the social team and what issues this team were working with in the region. The fourth interview provided an in-depth understanding into the wage management program, being one of the issues that the social team in Bangladesh worked with. The fifth interview gave an in-depth understanding on the water project, being one of the issues of focus within the environmental team.

**Table 1.** H&M interview data.

| Interviewees at H&M | Date | HQ Level /Bangladesh Market Region | Business Area in Focus | Length of Interview | Digital/ Physical Meeting |
| --- | --- | --- | --- | --- | --- |
| Sustainability manager H&M Nordics | 20191106 | HQ | Overall organization of sustainable activities | 66 min | Physical |
| Strategy Lead, Climate and Water, H&M Group | 20191217 | HQ | Environmental team: Climate and water strategies | 60 min | Physical |
| Global sustainability steering and development manager, H&M Group. | 20200131 | HQ | Global sustainability department; vision and strategy | 29 min | Physical |
| Strategy Lead, Fair Jobs Strategy, H&M Group | 20200220 | HQ | Social team: Fair jobs strategy | 49 min | Physical |
| Sustainability manager H&M Nordics | 20200310 | HQ | Sustainability team in Bangladesh: organization | 45 min | Physical |
| Sustainability team in Bangladesh, H&M Group: (1) Sustainability manager (2) Social program manager (3) Environmental program Manager | 20200618 | EM focus: Bangladesh | Sustainability team in Bangladesh: organization and activities. | 49 min | Digital |
| Sustainability team in the Bangladesh region, H&M Group: (1) Environmental program manager (2) Social program manager (3) Sustainability manager | 20200828 | EM focus: Bangladesh | Environmental team in Bangladesh: organization and activities | 54 min | Digital |
| Sustainability team in the Bangladesh region, H&M Group: (1) Social program manager (2) Environmental program manager (3) Sustainability manager | 20200925 | EM focus: Bangladesh | Social team in Bangladesh: organization and activities | 59 min. | Digital |
| Sustainability team in the Bangladesh region, H&M Group: (1) Sustainability developers (two managers) (2) Social program manager (3) Sustainability manager | 20201015 | EM focus: Bangladesh | Wage management program in Bangladesh | 51 min | Digital |
| Sustainability team in the Bangladesh region, H&M Group: (1) Environmental program manager (2) Sustainability program manager (3) Social program manager (4) Sustainability manager | 20201211 | EM focus: Bangladesh | Water project: organization and activities | 51 min | Digital |

To ensure the validity of our interview questions, we used CSR and sustainability documents, as well as news articles and reports from the business press, as anchor points for

the questions [53–56]. Various documents presenting and explaining the MNE's sustainability approaches have offered deep and detailed knowledge regarding sustainability work, including knowledge on how sustainability activities are organized and implemented, relationships with different stakeholders, different ethical codes, commitments, etc.

To enhance the reliability of our data collection, the interviews were transcribed and analyzed together with the documents using the conceptually clustered matrix detailed coding techniques and pattern matching recommended by Miles and Huberman [57]. The NVIVO 12 software package was used to catalogue, collect, and sort both the interview transcripts and the secondary data sources (e.g., annual reports, CSR, and sustainability documents). When coding the data, we simultaneously iterated between data collection and analysis [58].

To make conceptual categorizations based on the data, we used an open coding strategy [58] anchored in our purpose, with proactive CS activities and sustainability innovations being the focal units of the analysis. In this process, we used narrative analysis techniques, which included identifying commonalities and differences and exploring recurring themes and patterns. In line with O'Dwyer [59], the data analysis was based on an iterative approach and was made in parallel with data collection. This enabled us to probe emerging themes in subsequent interviews. The initial list received from our open coding procedure generated attributes related to the organization of sustainability work in the production market of Bangladesh and attributes related to the wage management system. When coding the data, one category focused on the specific challenges related to wages and the implementation of the wage management system. One category looked into the innovations and solutions to those challenges experienced, and here, we coded the data based on what level (i.e., factory level or industry level) the activities were undertaken. Some of the activities clearly related to the factories, and other innovative activities were focused on the whole fashion industry. Hence, the coding process generated attributes related to specific market-shaping activities on the factory and industry levels, and we could link them to innovations.

## 4. Results

### 4.1. Empirical Case of H&M in Bangladesh

H&M was established in 1947, then as a single store for women's clothing called Hennes located in Västerås, Sweden. Today, H&M has developed into a global fashion retailer, with presence in just over 70 retail markets and 21 production markets. The H&M Group employs over 180,000, and the group reportedly contributes to 1.6 million jobs in its supply chain. H&M is a fast fashion giant whose business model builds on short production cycles and quick time to market. As such, H&M has been frequently criticized for driving consumption and favouring (low) costs over quality and thereby exploiting labour in EMs by paying low wages and offering poor work conditions [60]. As stated in the Sustainability Report [54], H&M's "vision is to lead the change towards a circular and renewable fashion industry, while being a fair and equal company. Using our size and scale, we are working to catalyse systemic changes across our own operations, our entire value chain and the wider industry." Thus, sustainability is integrated in all business processes to weed out socially and environmentally harmful activities. The value proposition to consumers—"Fashion and quality at the best price in a sustainable way" or the "H&M way" implies that it is offered in the "ethical, honest, and responsible way" (the H&M way). What is ethical and sustainable is, of course, debatable; critics could claim that H&M's fast fashion value offering and the company's aspirations of being a sustainable company is a contradiction in terms [61].

Ever since the company was established in 1947, the H&M Group has outsourced its production. H&M has subcontracted operations to locally owned or multinational garment manufacturers, based mainly in Asia and Europe. Compared to many of the competing brands in the production markets, H&M believes it takes on a rather unique position. As one respondent points out: "We are pretty much alone in being physically present in

the production market". This presence is manifested by a company-owned production office located in each production market in 2019; the company operates 21 production offices situated in Europe, Asia, and Ethiopia. The 21 production offices, in turn, support around 750 suppliers on a regular basis. One respondent at the HQ pointed out that H&M's competitors prefer to work with agents or other parties that, in turn, handle the relations with local suppliers. Hence, being physically present at each production market and closely collaborating with suppliers in the local markets is something that distinguishes H&M from its competitors.

Much of H&Ms production is in countries where labour laws and labour market institutions are often missing. In cases when they do exist, they may be underdeveloped or do not conform to international standards. For H&M, this implies that the company needs to closely monitor that work conditions comply with the code of conduct regarding issues such as living wages, worker representation, and development opportunities. In this work, H&M focused on building strong, long-term relationships with its local stakeholders. The respondents emphasized that a physical presence enables H&M to work more actively with sustainability issues in the production markets. It facilitates dialogues with suppliers and enables H&M to better understand the local market context that could shape the adoption of sustainability practices. Managers learn about recurring problems, culturally determined behaviours, and conflicts related to complying with sustainability principles.

Bangladesh is one of H&M's largest production markets. The H&M Group has 214 direct suppliers in Bangladesh; some of these suppliers in Bangladesh have collaborated with H&M for 20 years. In the production office in Bangladesh, the local sustainability team consists of 28 people. They work solely on sustainability issues in this region, organized into an environmental team—focusing on issues related to water and energy use—and one social team—focusing mainly on issues related to living wages, industrial relations, and skills development. While H&M works actively in all these areas, we will, for the purpose of stringency, focus on challenges concerning living wages and the ways in which the company has tackled these challenges with a (more or less) proactive approach to CS.

### 4.2. Grand Challenge—To Ensure Good Working Conditions and Improved Wages in the Market with Underdeveloped Labour Laws

Fair wages are one of the United Nations Sustainable Development Goals. H&M has created a roadmap to reach this objective: "Every garment worker should earn enough to live on"—the initial implementation of the Fair Living Wage Roadmap. However, ensuring living wages for workers in supplier factories has been challenging. The commitment to implement living wages was made in 2013, but the company has yet not delivered on the promise. H&M has received massive criticism from the media and civil society groups for not paying living wages, forcing employees to work excessive hours [60].

Making sure garment workers are paid fairly is a big challenge in the Bangladesh region, as well as in many other production markets. First, since H&M does not own or manage the factories, it means the company does not pay garment workers' salaries and can therefore not decide how much they are paid. Secondly, in many markets, workers have limited possibilities to negotiate wages collectively using union representatives. In addition to these challenges, H&M often faces the situation that factories are normally contracted by many different brands. H&M needs to find innovative ways to work around these challenges and still ensure that individual factories can pay fair living wages to their workers. Adding to these challenges, H&M emphasizes that income equality also needs to be seen in relation to the other sectors [62]. For example, the company acknowledges that they need to consider the consequences that might arise for a community where garment workers start to earn more than teachers might [62].

### 4.3. Proactive CS at Market Level—Fair Living Wage Roadmap

The Fair Living Wage Roadmap was launched in 2013 as a strategy to reach the mission to guarantee a fair living wage for every garment worker. The roadmap was to guide H&M

in understanding and improving workplace dialogues and wage management systems at the factory level and ensuring appropriate purchasing practices and collaboration with other stakeholders in the industry. To reach the mission, the roadmap included not only H&M, the factory owners, and factory employees but also engaging concerned governments on wage issues [62]. In order to achieve fair living wages for all garment workers, it is important to understand the components of a textile worker's monthly take home wage and how they can be influenced. In general, there are two components: The biggest component is minimum wages, which are stipulated by governments, and the lesser component is individual wage settings and factory benefits. Combined, these components make up the take home wages for garment workers. To engage governments to support higher wages for garment workers is thus one of the focus areas in the roadmap.

Already, in 2013, H&M was in contact with the Bangladeshi government and requested increases in minimum wages and systematic minimum wage revision [62]. In 2014 and 2015, H&M started to pilot the implementation of the fair wage method in some carefully selected factories in Bangladesh. The aim was to highlight that paying a fair living wage can go well together with the best business performances. From there, H&M were to gradually scale up the implementation. In starting to talk with some of the first factories in Bangladesh on the issue of wages, the local team experienced major pushback from the factories, where managers could not see the business logic of raising wages for no apparent reason. One respondent involved in the development and initial implementation of the fair wage method recalled these initial talks with suppliers in Bangladesh:

> "I remember when we were to implement the strategy and I had meeting after meeting with individual suppliers. We did not want to gather too many suppliers in a meeting, because if one were negative, it could influence the other suppliers to become negative as well. So, with individual meetings we started to explain what we wanted to achieve and what we wanted to improve related to wage systems, having individual wages. We experienced suppliers that turned to us and said, "Why are you a buyer that wants to have a dialogue about wages?" They said they did not want to be part of such a dialogue. In this situation, it was really important for us to find those suppliers that were willing to listen."

The respondent explained how it became very clear in this initial implementation phase that most suppliers were not used to having a dialogue about wages with their buyers. In 2013 and 2014, there were no other buyers who placed such requirements on their suppliers. It became even more difficult to follow through with this initiative, because H&M had limited experience with engaging suppliers in a constructive discussion about wages. One respondent explained the dilemma H&M managers experienced in trying to shift the perspective of suppliers:

> "We tried to convince suppliers that they will see an increased motivation among their workers. But we could not say that this had happened in Bangladesh, because there was yet no proof of this effect."

Looking back at the initial implementation phase, the respondent described how introducing and implementing the fair living wage model is an example of something that "we, as a Swedish based multinational firm, believe in and are convinced about, but to the local suppliers the fair living wage model has been very unusual and difficult to understand."

*4.4. Proactive CS at Factory Level—Changing the Suppliers' Mindsets*

The respondent working with the fair wage method at the HQ explained that, when working with social issues such as wages, progress is sometimes immeasurable. The company has to rely on soft key performance indicators (KPI). It concerns improvements of an intangible nature, such as increased perceived work satisfaction, increased perceived motivation among workers, and increased perceived productivity. Environmental issues,

in contrast, are often easier to measure, and the KPI can be linked to tangible scores related to, e.g., water usage in litres and carbon dioxide emissions in tonnes.

The social local team in Bangladesh working on the living wage issues described in detail the challenges connected to working with soft KPIs in the dialogues with the local factories and suppliers. When engaging suppliers in a dialogue around wage practices, one would perhaps expect resistance about the costs involved. However, the major pushback in this discussion was not associated with a possible increase in the wage level. The H&M managers, at both the HQ and the social team in Bangladesh, claimed that it was changing the mindsets of factory managers related to disclosure practices of wage information that was the tricky part. Factory workers have to relate to new benchmarks of which they have no prior experience. A proper wage management system is reflected by the transparency of evaluation parameters and the accuracy of the evaluation system. When a worker is evaluated in a transparent and accurate manner, the workers can see what she or he needs to do to improve to go step up to the next level. Based on that understanding, the workers should be more motivated. A transparent system reduces room for nepotism, corruption, and biased decisions. One respondent from the social team in Bangladesh pointed out "It's all about the acceptance from the management level." In addition, another respondent highlighted that "it takes a long time before our suppliers change their mind-set on these issues."

In the process of implementing the wage management system and initiating a dialogue with factories, the respondents could see that, once the initial big factories began to participate in this dialogue, the discussion attracted the interest of more factory managers. Managers in the factories realized that treating employees more fairly could be a competitive factor. If one supplier was beginning to appear more attractive, they could start to attract and recruit the employees of their competitors. Hence, managers in factories realized that a new wage system could increase employee satisfaction and lead to less turnover among the workers. In other words, the soft KPIs related to the living wage program could, over time, be substantiated and complemented by more tangible KPIs capturing employee retention.

Four to five years after the initial implementation, H&M have experienced how supplier managers have come back, recalling how they initially scorned the very premise of the wage program but now explaining that their attitudes have changed. Several managers do see the upside of working living wages and a transparent evaluation system. One H&M manager talked about "a change of mindset among several suppliers" over the years. This is particularly the case among those suppliers where the next young generation of managers have taken over. The younger generation enters in with a new and more open mindset related to sustainability issues.

*4.5. Proactive CS at the Industry Level—Collaboration with Other Global Companies*

In an interview with the CEO of H&M [62], the CEO pointed out that, to create a sustainable fashion industry, lasting and systemic change cannot be made by one company alone. While working together with others may take more time, collaboration with several different stakeholders and partners is the key to lasting change from the perspective of H&M. A key issue in taking the wage issue to the next level within the industry is, thus, to engage concerned stakeholders, as well as governments, to promote systemic change. One example of a crucial collaboration for H&M in this endeavour is the formation of ACT.

In 2015, H&M joined forces with the global union organization IndustriALL and several other brands to form the collaboration called ACT (Action, Collaboration, and Transformation). The mission of ACT is to transform the garment, textile, and footwear industry and achieve living wages for workers through collective bargaining at the industry level. ACT is the first global commitment on living wages in the sector that provides a framework through which all relevant actors, including brands and retailers, trade unions, manufacturers, and governments, can exercise their responsibility and role in achieving living wages. ACT was founded by 17 global brands and retailers, including H&M Group, and IndustriALL Global Union, who represent garment and textile workers from around the globe. Today, ACT includes 22 brands. For H&M, the formation of ACT represents

an important step on the journey to fair wages. One respondent explained that, of course, their arguments towards suppliers and governments become stronger when coming from 22 companies. Hence, the company's leverage towards policy-makers in the production countries is gaining increased traction and describes ACT as "an innovative solution, in terms of being the first time 22 companies in the industry go together and agree on improvements regarding issues related to living wages".

*4.6. Two Phases of Social Innovation Implementation*

Based on the examples of proactive CS at the factory and industry levels, we identified two phases of social innovation implementation: the ideation phase and testing/modification phase. In the ideation phase, represented by the factory level activities Fair Living Wage Roadmap (see Section 4.3) and changing the suppliers' mindsets (see Section 4.4), the preparatory work is being carried out to lay the foundation for the testing/modification phase, represented by industry-level activities (see Section 4.5). In the ideation phase, a broad behavioural change is taking place among the multiple stakeholders involved, while, in the testing/modification phase, structural changes are being prepared to gradually take place. We will discuss the critical factors pertinent to each phase in the next section.

## 5. Discussion

The case of H&M demonstrates how the company attempts to address deeply rooted social problems through mission-oriented social innovation via a number of critical market-shaping activities, ultimately leading to the change of market actors' behaviours and the market structure in an important production market and, in the process, tackling a grand challenge. While the literature recognizes the multifaceted nature of social innovation [41], captured here by task-oriented social innovation (limited in scope and local/situation-specific) and mission-oriented social innovation (potential wide-reaching societal impact), H&M is seemingly inclined towards mission-oriented innovation. A plausible reason for this is the magnitude of its business and the nature of its business model, where there are clear negative externalities associated with its operations. These negative externalities relate to issues that are widely recognized by civil society and can be linked to several of the UN's Sustainable Development Goals. Reducing damage and pursuing minimal sustainability standards may simply not be enough to please stakeholders for H&M. To be viewed as credible and to retain the brand, H&M has formulated high set goals for its operations that go beyond fixing problems in markets and aims to reshape markets [17,48]. To offer a far-reaching value proposition [14,18] and proactively enhance their reputation [25], the introduction of ambitious social innovations programs has been instrumental.

The case examined in this study, striving for living wages for employees at production sites in Bangladesh, shows that reaching high set objectives to grand challenges is indeed not easy. H&M has notoriously failed to meet its targets when attempting to implement living wages in subcontractor factories. Nevertheless, the company tried to change mindsets and practices among factory managers and workers by devising wage programs aimed at increased wage levels, performance-based wages, and transparent wages.

*5.1. The Implementation of Mission-Oriented Social Innovation*

Guided by the conceptual framework developed in Figure 1, the case provides some in-depth illustrations and insights on the implementation of mission-oriented social innovation and market-shaping activities of importance. Figure 2 summarizes the key findings from the case analysis. The case clearly shows that the launch and development of social innovation programs is not a streamlined process. We identified two phases of social innovation implementation: the ideation phase and testing/modification phase. In the ideation phase, in line with Elg et al. [31], we found that, to implement mission-oriented social innovation aimed at changing ingrained social behaviours, the company needs to engage stakeholders (i.e., competing brands, workers, unions, NGOs, and factories). This involves making sure that participants see the benefits of changing wage practices and to develop

solutions that resonate. Hence, in line with Nylund et al. [45], we found that stakeholder engagement thus seems to play a crucial role in initiating and managing the implementation of social innovations. The case also showed that the implementation was a learning process, where new insights generated along the way perpetuated the development of the social innovation regime. H&M gradually learned about managers' points of views and, based on their acquired market knowledge, constructed arguments that were more effective and created practices that targeted problematic behaviours. Mission-oriented innovation, moreover, requires the concerted action of a variety of stakeholders [48]. H&M seemed to take on a leading role in orchestrating stakeholders such as factory managers, workers, NGOs, and other global companies. H&M assumed the role as a thought leader and the source for the ideation of new practices and for the evaluation of those practices post-implementation in the testing/modification phase. While the company's legitimacy in this role appeared weak at the outset, it seems to have been reinforced, as factory managers have begun to see results of their efforts. This has been achieved by testing new solutions and strengthening the social value proposition via the best practices shared among the suppliers. H&M has dealt with regular setbacks in reaching the objectives related to living wages, which have made the company a target for negative publicity and criticism despite the importance given to relationship-building activities with the stakeholders. In response, H&M has intensified their activities related to social innovations within the wage programs in production markets and are continually revising the plans for how to expedite these changes. In this phase, via sharing the best practices with industry participants through ACT collaboration, H&M has made big steps towards the introduction of new standards in the garment, textile, and footwear industry.

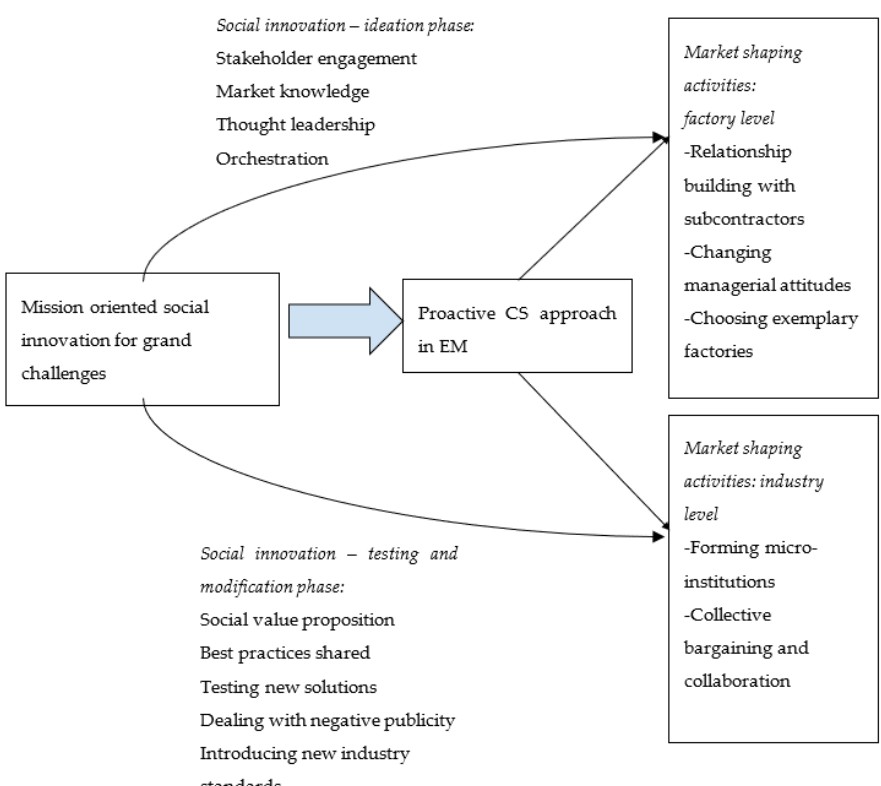

**Figure 2.** Findings from the case study.

We also identified two levels at which the company designs specific activities to shape the behaviours of critical stakeholders and change the industry structure for the aim of implementing living wages: the factory level and the industry level. On the factory level, it has been necessary for H&M to maintain close relationships with the subcontractors

producing the company's garments in their factories. Through relationship-building activities with subcontractors, H&M has been able to exert influence and gradually shape the mindsets of key individuals. As the idea of wage management practices gained increasing footholds, a cadre of exemplary factories emerged for others to benchmark against. In this way, new and different managerial and worker behaviours were created at the factory level. Changing attitudes towards more transparency and dialogue at the factory level coevolved through concerted action orchestrated by H&M at the industry level. By joining forces with competing brands, new micro-institutions were created, equipped with functionalities that facilitated knowledge diffusion and paved the way for managers and workers to organize new wage practices. This was clearly illustrated in the formation of ACT, where H&M took on a proactive role already in its early formation. This level of proactive CS was aimed at changing the industry standards and structure through collective bargaining and agreements among the leading players in the industry.

### 5.2. Social Innovations as Means for Market Shaping and Proactive CS

Building on our conceptual framework and Figure 2, the empirical findings also helped us advance our theoretical discussion on how social innovations can serve as important means for proactive CS in MNE's operations in an EM. First, the role of social innovation for market shaping seems to differ from business innovation (tangible solutions) [18], since the latter have to be put in place before any reordering of market activities can take place. The case analysis shows how social innovation is modified during the process of market shaping: It starts from articulating missions and strategic intent (ideation phase) and grows into the specific set of activities and solutions that are being tested in the market for whether they work or not (testing and modification phase). The role of social innovation is thus to transform the existing norms and behaviours into realistic solutions accepted by local stakeholders, which implies a long-time horizon.

Looking into the market-shaping activities being undertaken, we discern similarities, as well as differences, in relation to the previous literature on the subject. The market-shaping literature identifies a broad range of market-shaping activities and acknowledges that these can be pursued at different levels, e.g., the system level, market offer level, and technology level [31,32]. The industry level, emphasized in the case analysis of H&M, is similar to what the literature describes as the system level [32]. Like system level influences, the industry level influence demonstrated in the H&M case focused on norms and standards that set rules for the entire fashion industry and not only competitors, but the potential actors involved in the larger production market of Bangladesh. Hence, the activities identified in this case illustrate the development of new institutional arrangements that can permeate the entire market. It is also possible to draw a line between the factory levels identified in the case analysis to what the literature identifies as the market offer level [32]. However, in this study, we do not focus upon a market offer in terms of products or services. The social offer is a potential value proposition inherent in implementing wage management systems that provide the workers with not only "fair" wages but also increased transparency and dialogue between workers and factory managers. On the factory level, we can see activities driving the formation of a value proposition around novel practices in wage management systems. These activities and value propositions are clearly emerging out of close interactions and relationships with subcontractors and are driven by the vision of leading changes in the industry towards a more sustainable value chain. This social value proposition is aimed at the increased values (benefits) for workers and the means to create lasting changes in the industry, thus being similar to what the literature calls the "leap in value proposition" to customers [18] but in a broader and more inclusive sense.

On a final note, while the literature [32] argues for the need of synergetic activities on all three levels: system, market, and technology, with technology (business innovation) often functioning as a trigger for learning and change at the early stages of the market-shaping process, in proactive CS, social innovation does not constitute a separate level of

influence. Rather, it functions as an overall frame and blueprint for action, linking the two levels of industry and factory activities in their transformation towards higher standards and inclusiveness in the global value chain. Due to its dynamic nature of transforming the existing norms and being transformed into broadly accepted solutions, social innovation represents an ongoing project involving multiple stakeholders, which requires a long-term commitment from MNEs. Seen in this light, the company's failures to meet its targets when implementing social innovation can be seen as important steps towards its maturity.

## 6. Conclusions

We set out to explore how MNEs can use social innovations to drive change and shape market behaviours and market structures in an EM context. We found that mission-oriented social innovation regimes can influence underlying market structures and thereby remedy problems that are constituents of imperfect markets. This proactive approach to CS does not only mitigate risk and control damage; it can also facilitate business exchange. The study shows that social innovation can function as an engine for change in EM settings. Through mission-oriented social innovation initiatives, MNEs can become orchestrators, leveraging resources both at the factory level and at the industry level to introduce new practices, norms, and micro-institutions that can govern future transactions in line with the CS principles. However, it requires true commitment on behalf of the firm and a willingness to endure public scrutiny to engage in social innovation.

As discussed in the analysis section, we contributed to IB literature by theorizing about the role of social innovations as important means to drive changes and push the CS agendas of MNEs. While social innovations have been treated quite vaguely in the extant literature, we outlined a specific type of social innovation aimed at grand challenges and described how this focus can be an important catalyst for proactive CS activities. We were also able to empirically substantiate how mission-oriented innovation regimes are implemented and how they can tackle grand challenges. This study represents an early attempt to delve into the underpinnings of the actual implementation of mission-oriented innovation in EM settings by one MNC, and this way is limited in terms of generalizability to other MNE CS implementations in EMs. However, we suggested that some key analytical insights might be relevant for MNEs in general. For example, we found that social processes (the ideation phase and the testing/modification phase) are instrumental in implementing these social innovation regimes. Based on this study, we argued that frameworks focusing on risk mitigation and compliance do not account for the full picture to understand MNE sustainability activities in EMs. MNEs may need to act proactively in relation to social problems to bring about meaningful changes.

Based on this study, we call for further research and conceptualizations that consider the modes of governance around social innovations, e.g., ecosystems that emerge around social innovations in various settings. A topic related to social innovation in ecosystems worth further scholarly attention is the notion of orchestration. How can MNEs orchestrate changes in ecosystem settings? What external circumstances and internal organizational assets are favourable for orchestration? We also encourage researchers to delve deeper into the issue of market conflict that could surface because of launching social innovations in market settings where MNEs' sustainability principles clash with the local norms (which we saw some evidence of in this study).

This study comes with managerial implications. While mission-oriented social innovation regimes have great potential to connect more closely with local communities and to develop corporate brands, the outcomes of these projects are notoriously difficult to measure. We therefore encourage managers to not solely focus on grand objectives (e.g., zero carbon emission) but also set up achievable milestones. We recommend that managers create dual incentives for stakeholders to pick up new practices, i.e., producing credible evidence of both economic incentives and social incentives to expedite participation.

**Author Contributions:** Conceptualization, V.T. and D.T.; methodology, S.M.H.; software, V.T. and S.M.H.; formal analysis, V.T., D.T. and S.M.H.; writing—original draft preparation, V.T., D.T. and S.M.H.; and writing—review and editing, V.T., D.T. and S.M.H. All authors have read and agreed to the published version of the manuscript.

**Funding:** This research was funded by Riksbankens Jubileumsfond.

**Institutional Review Board Statement:** Not applicable.

**Informed Consent Statement:** Informed consent was obtained from all subjects involved in the study.

**Data Availability Statement:** Not applicable.

**Conflicts of Interest:** The authors declare no conflict of interest.

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
