# Peer review of "Proactive Corporate Sustainability via Social Innovation—A Case Study of the Hennes & Mauritz Grand Challenge in Bangladesh"

_sustainability, doi:10.3390/su14020599_

Round 1

Reviewer 1 Report

Dear authors,

congratulations for the work presented.

I read your article with great interest, and I give you some suggestions so that you can improve this work a little more.

Abstract and keywords:

-Abbreviations should not be used when the reader is still unaware of their meaning (“MNE”, “H & M's”, “CS”).

Introduction and Theoretical Framework:

- The introduction places the study in a broader context and highlights why it is important and defines the purpose of the work. However, it would be interesting for the authors to present some arguments why, in practice, companies have such low levels of SC, in order to highlight the controversy or divergences that make the study even more pertinent.

Methodology:

- Properly and correctly applied. The authors disclose how the investigation was conducted, in sufficient detail so that it can be replicated. However, it is seriously good that the analysis documents addressed in each interview conducted and which documents were consulted were disclosed. I suggest that table 1 can be reinforced with this information.

Analysis and Discussion of Results:

- The authors present and discuss the results that come to the present study, suitable for calculating the defined objective. I would like to highlight Figure 2, which summarizes the contributions of the article very well.

Conclusions:

- It would be good for the authors to present clues for future investigations.

References:

- References related to following the guidelines of the journal, must contain the DOI and some are poorly pointed (Example, ref. Nr. 58).

Good luck

Author Response

Proactive Corporate Sustainability via Social Innovation – a Case Study of Hennes & Mauritz’ Grand Challenge in Bangladesh

Letter to reviewers round 1

Dear reviewer!

Thank you for providing us with the valuable comments and suggestion on how to improve our paper. We have taken all your suggestions into consideration and hereafter address each reviewer separately.

Answer to reviewer 1

Thank you very much for your positive response and specific comments. We have addressed them in the following way:

1)Abbreviations should not be used when the reader is still unaware of their meaning (“MNE”, “H & M's”, “CS”).

Our response: We have provided full names and titles to all abbreviations the first time they are mentioned. Examples (in the Introduction) - Research has pointed at an increasing number of MNEs that, like H&M, are treating corporate sustainability (CS) as means to gain a competitive edge. Multinational Enterprises (MNEs) are to an increasing extent expected to behave… One such example is the Swedish fast-fashion company Hennes & Mauritz (H&M).

2) The introduction places the study in a broader context and highlights why it is important and defines the purpose of the work. However, it would be interesting for the authors to present some arguments why, in practice, companies have such low levels of SC, in order to highlight the controversy or divergences that make the study even more pertinent.

Our response: Addressing this point, we have stressed the relevance of CS but also pointed at lack of insight about this subject. In this way we, in line with your comment, justify the reason to study this issue.  For example, we argued although CS “has become a strategic priority for the company to avoid reputational damage but also to develop the brand.” CS has not only be viewed as a reactive activity, though, meaning that there is some controversy there: “MNEs that, like H&M, are treating corporate sustainability (CS) as means to gain a competitive edge (e.g. Park and Ghauri, 2015; O´Rourke and Strand, 2017; Schaltegger and Burritt, 2018; Burritt et al. 2020). (see Introduction). With this debate as a backdrop, we claim that we have a limited knowledge of how companies use social innovation as part of their CS - “While MNEs may have the resources and legitimacy to drive social innovation programmes that have an impact on developing societies, this issue remains largely unexplored. In comparison to corporate social strategies which outline a framework for minimum levels of compliance, social innovation programmes go beyond that and may lead to more progressive outcomes. We think this topic deserves more attention since combining public and private efforts to create social change may be a way to solve the grand challenges of the 21st century.” (see Introduction)

3) Methodology: The authors disclose how the investigation was conducted, in sufficient detail so that it can be replicated. However, it is seriously good that the analysis documents addressed in each interview conducted and which documents were consulted were disclosed. I suggest that table 1 can be reinforced with this information.

Our response: To ensure the validity of our interview questions, we used CSR and Sustainability documents as well as news articles and reports from the business press, as anchor points for the questions. We try to cleary specify when we use specific sources (e.g., H&M 2018a; H&M 2018b; H&M 2019a; H&M 2019b).

4) Analysis and Discussion of Results: the authors present and discuss the results that come to the present study, suitable for calculating the defined objective. I would like to highlight Figure 2, which summarizes the contributions of the article very well.

Our response: Thank you very much for a very positive comment!

5) Conclusions: It would be good for the authors to present clues for future investigations.

Our response: Again, a very good suggestion which enables us to make a stronger contribution. We have added these aspects to the “Conclusion” section:

Based on this study, we call for further research and conceptualizations that take into account the modes of governance around social innovations, e.g. ecosystems that emerge around social innovations in various settings. A topic related to social innovation in ecosystems worth further scholarly attention is the notion of orchestration. How can MNEs orchestrate change in ecosystem settings? What external circumstances and internal organizational assets are favorable for orchestration? We also encourage researchers to delve deeper into the issue of market conflict that could surface because of launching social innovations in market settings where MNEs’ sustainability principles clash with local norms (which we have seen some evidence of in this study). 

6) References: References related to following the guidelines of the journal, must contain the DOI and some are poorly pointed (Example, ref. Nr. 58).

Our response: Thank you for drawing our attention to this aspect. We will do our best this time and make sure we have all DOI in place.

Reviewer 2 Report

The manuscript Proactive Corporate Sustainability via Social Innovation – a Case Study of Hennes & Mauritz’s Grand Challenge in Bangladesh deals with a relevant topic. It seems that H&M invests and allocates time and resources to social problems and to bring an improvement in quality of life through living wages. These activities are likely to be conducted under the enterprise’s corporate social responsibility strategy rather than an SI strategy. Thus, further explanation is needed to eliminate this confusion. Furthermore, more recent literature should be considered and added to this manuscript. Moreover, additional quotations from interviews would improve the quality of the manuscript.

Otherwise, the paper is well written, provides interesting insights into how H&M is dealing with issues related to living wages in Bangladesh.

Minor issues:

  • References are incomplete.
  • The explanation of the abbreviation EM is missing. Emerging market?

Author Response

Proactive Corporate Sustainability via Social Innovation – a Case Study of Hennes & Mauritz’ Grand Challenge in Bangladesh

Letter to reviewers round 1

Dear reviewer!

Thank you for providing us with the valuable comments and suggestion on how to improve our paper. We have taken all your suggestions into consideration and hereafter address each reviewer separately.

Answer to reviewer 2

Our response: Thank you very much for your positive response and specific comments. We have addressed them in the following way:

  • The manuscript Proactive Corporate Sustainability via Social Innovation – a Case Study of Hennes & Mauritz’s Grand Challenge in Bangladesh deals with a relevant topic. It seems that H&M invests and allocates time and resources to social problems and to bring an improvement in quality of life through living wages. These activities are likely to be conducted under the enterprise’s corporate social responsibility strategy rather than an SI strategy. Thus, further explanation is needed to eliminate this confusion. Furthermore, more recent literature should be considered and added to this manuscript. Moreover, additional quotations from interviews would improve the quality of the manuscript.

Our response: Thank you for a positive response to our paper! You make a good point saying that we need to clarify the relationship between Corporate Sustainability and Social Innovation.  We consider now try to be clearer in conveying that we consider social innovation as an integral part of CSR /CS strategy.

”While MNEs may have the resources and legitimacy to drive social innovation programmes that have an impact on developing societies, this issue remains largely unexplored. In comparison to corporate social strategies which outline a framework for minimum levels of compliance, social innovation programmes go beyond that and may lead to more progressive outcomes. We think this topic deserves more attention since combining public and private efforts to create social change may be a way to solve the grand challenges of the 21st century.”

We have also tried to add another quote in the case, and more recent literature to our manuscript. See for example reference 45 and 46:

  1. Nylund, P. A.; Brem, A.; Agarwal, N. Innovation ecosystems for meeting sustainable development goals: The evolving roles of multinational enterprises. Journal of Cleaner Production 2021, 281, 125329.
  2. Prashantham, S.; Birkinshaw, J. MNE–SME cooperation: An integrative framework. Journal of International Business Studies 2020 51, 1161-1175.

Minor issues:

  • References are incomplete.
  • The explanation of the abbreviation EM is missing. Emerging market?

Our response: Thank you for pointing this out (EM = emerging market, which is now clarified). We have added the missing references and explained all abbreviations in the text.

Reviewer 3 Report

Thank you for submitting your manuscript to the Sustainability journal.
 Generally, the manuscript fits into the journal's scope. However, the manuscript has major flaws. 
1. The abstract should define clearly the purpose of the study, the methods used, the key results and the study conclusions or interpretations.
2. In the introductory section, there is a need to clarify the study aim or focus. It was stated in lines 58-60 that "In this article, we will bridge the sustainability and market-shaping driving literature to study the implementation of proactive CS by an MNE with a specific focus on social innovation". Likewise, in lines 73-75, it was also stated that "the purpose of this study is to explore how MNE can use social innovations to drive change and solve grand challenges in an EM context". These are conflicting statements. 
3. The literature review/theoretical framework section is poor and needs to be upgraded. In the literature review, the scientific novelty of the work must be established through a critical analysis of related literature. How does this work contribute to the gaps identified? How does it improve upon previous work? Avoid overly being descriptive. Besides, it is unclear how the review conducted in section 2 theoretical framework gives rise to Figure 1. Conceptual framework of how social innovation can be a means of proactive CS in an emerging market context. The section has not provided any relevant review or critique of the literature supporting the relationship between the different variables within the proposed framework. I would have expected to see the study establish or critique the relationship between sustainability, market-shaping and social innovation.

4. The method section is fairly ok but descriptive. 
5. The analysis of the study result is poorly presented, descriptive, and this further makes it difficult for the audience to pinpoint the study findings. In addition, there is no link between the study analysis (i.e. section 5) and the discussion (i.e. section 6). It is unclear where the findings discussed in section 6 emerged from. 
6. There is a need to provide clear justification as to how this study extends or contributes to sustainability or social innovation. 
7. Abbreviations should be stated in full when first used within the manuscript — for example, H&M and EM. 
8. Most references are dated. Consider using more recent references. 

Author Response

Proactive Corporate Sustainability via Social Innovation – a Case Study of Hennes & Mauritz’ Grand Challenge in Bangladesh

Letter to reviewers round 1

Dear reviewer!

Thank you for providing us with the valuable comments and suggestion on how to improve our paper. We have taken all your suggestions into consideration and hereafter address each reviewer separately.

Answer to reviewer 3.

Thank you very much for your criticism and specific comments. We have addressed them in the following way:

  1. The abstract should clearly define the purpose of the study, the methods used, the key results and the study conclusions or interpretations.

Our response: Thanks for pointing this out. We have re-written the abstract according to your suggestions.

“The purpose of the study is to explore how a multinational enterprise can use social innovations to drive change and solve grand challenges in an emerging market context. The paper brings market-shaping literature into a sustainability context, particularly by studying the implementation of social innovations in an emerging market context. Specifically, the study. involves an in-depth qualitative study of H&M’s fair living wages program in Bangladesh. We find that H&M is tackling utterances of grand challenges revealed by orchestrating social innovation in collaborations with local stakeholders. Social innovation is carried out in on-going projects involving multiple stakeholders. The study contributes to current literature by revealing that multinational enterprises indeed can use social innovation to drive change in emerging markets, although this requires long-term commitment, an ability and willingness to shape the surrounding business environment, and a prominent standing among key stakeholders”.

  1. In the introductory section, there is a need to clarify the study aim or focus. It was stated in lines 58-60 that "In this article, we will bridge the sustainability and market-shaping driving literature to study the implementation of proactive CS by an MNE with a specific focus on social innovation". Likewise, in lines 73-75, it was also stated that "the purpose of this study is to explore how MNE can use social innovations to drive change and solve grand challenges in an EM context". These are conflicting statements. 

Our response: Thank you for spotting this. We have clarified this confusion by clearly stating that: 

“In this article, we bring market-shaping literature into a sustainability context, particularly by studying the implementation of social innovations in an emerging market (EM) context.”

This sentence can now, thus, not be confused with the purpose but merely conveys the literature the study is based on. The purpose, furthermore, is slightly modified to:

 “The purpose of this study is to explore how a MNE can use social innovations to drive change and solve grand challenges in an EM context”.

  1. The literature review/theoretical framework section is poor and needs to be upgraded. In the literature review, the scientific novelty of the work must be established through a critical analysis of related literature. How does this work contribute to the gaps identified? How does it improve upon previous work? Avoid overly being descriptive. Besides, it is unclear how the review conducted in section 2 theoretical framework gives rise to Figure 1. Conceptual framework of how social innovation can be a means of proactive CS in an emerging market context. The section has not provided any relevant review or critique of the literature supporting the relationship between the different variables within the proposed framework. I would have expected to see the study establish or critique the relationship between sustainability, market-shaping and social innovation.

Our response: Thank you for this challenging but very constructive comment. We have tried to make substantial improvements in the theoretical section to make the contribution clearer. In the paper, we have paved the way towards the relationship between sustainability, market shaping and innovation by:

  • arguing that there is a lack of studies on the proactive CS as a response to grand challenges on both conceptual and empirical levels. In section 2.1. we now say: “Although there is increasing awareness on the embeddedness of MNEs within their local, regional, and global context (Peng et al. 2008) and the nature of interactions among MNEs and external actors, studies unpacking the MNE’s proactive responses to local and global challenges are few. For example, Regner and Edman’s (2014) can identify factors enabling MNEs subunits to shape the local institutional contexts to their advantage, but the study still focuses on the focal firm and does not consider the broader societal perspective. Consequently, a deeper understanding of proactive CS as a market shaping approach might be a way to understand MNEs’ responsiveness to societal problems ingrained in local communities, in turn affecting economic life.  As argued in Buckley et al. (2017), it is vital that IB research establishes a stronger connection to the contextual characteristics of MNE’s environment to not only address economic performance, but also performance in relation to grand challenges. The empirical case study conducted in this study, featuring a specific grand challenge related to fair wages, adheres to such concerns. “
  • Introducing the review of market shaping literature and emphasizing the lack of understanding of the role of social innovation in the proactive CS. In section 2.2. we now say:

“Surprisingly, the role of social innovations as vehicles for market shaping has not been discussed in the marketing literature while given more attention in IB literature (e.g., Campbell et al. 2012). In this study, we are using the activities framework from the literature to uncover the specific activities carried out by MNC in the EM, with a special focus on the social innovations as enabling a firm to create lasting changes in market actors’ behaviours as well as structural changes in the industry.”

  • Explaining how the theoretical framework gives rise to Figure 1, see the following text in section 2.3:

“Recently, literature has drawn a line between social innovation and efforts to proactively tackle grand challenges, for example achieving UN’s social development goals (Nylund et al., 2021). Prashantham and Birkinshaw (2020) also argue that MNEs can benefit societies - and themselves - by cooperating with local companies in EMs where addressing SDGs is a pressing concern. Developing local relationships for this purpose would allow firms to combine complementary resources to address societal and economic needs simultaneously.  Correspondingly, Nylund et al. (2021) argue that to develop social innovations, MNEs form business ecosystems and networks involving local stakeholders that influence not only business but wider social systems. However, literature has not yet fully accounted for this diversity of social innovation, specifically related to the different types of social innovations as well as how social innovations can serve (or not serve) proactive efforts of CS.”

And, in this section we also point out the following:

“Furthermore, the role of social innovation in market shaping in general and in proactive CS as a strategic activity in particular has not been given adequate attention in the literature. By focusing on social innovation as means for proactive CS by MNEs in EM setting, we are answering multiple calls for phenomena-driven and broader interdisciplinary perspectives that suggest MNEs have a role to play in tackling grand challenges affecting societies at large. We propose, in turn, that a proactive market approach may involve moving beyond a risk mitigating focus related to CS issues and involve an inclination to tackle grand challenges by shaping markets to reap long-term benefits and sustainable change. In Figure 1, we combine the key tenets of market shaping research and social innovation for a preliminary framework for the study. The conceptual framework illustrates how social innovation (mission-oriented and task-oriented) can be used for proactive CS in an EM context, conceptualized as marketing shaping activities and a focus on grand challenge objective(s)”.

Finally, we have added this text to section 2.3.:

 “This framework is tentative in a sense that it helps us understand how social innovation manifests itself, how it is used by MNC to plan and implement different change-oriented activities aimed at local challenges and what the outcomes are. It does not imply the direct relationship between social innovation and proactive CS in a way that one leads to the other. We are suggesting that social innovation is an integral part of many MNCs CS strategies, but its role is not fully recognized. By specifically aiming at unpacking social innovation, we expect to contribute to CS literature in the field of international business by examining the role of social innovation as a driver of proactive CS strategies of MNEs.”

  1. The method section is fairly ok but descriptive.

Our response: We have added a paragraph of research limitations in the method section, and maybe it can strengthen the method section somewhat. We now say:

“We recognize that this single-case study of a firm in the fashion industry restricts us from generalizing results to other companies. For example, being a global consumer-oriented firm has certain implications on public scrutiny of CS operations, brand recognition, and social judgments in comparison with, for example, a business-to-business firm. Nonetheless, the quality and richness of data will allow us to capture a pertinent and emerging phenomenon and develop conceptualizations that could serve useful for future studies that seek to develop knowledge and theory about social innovation as a driver of CS development.  While consumer-oriented firms currently are seemingly taking the lead in this area, this is a growing phenomenon that over time could be applicable on a wider group of firms. “

  1. The analysis of the study result is poorly presented, descriptive, and this further makes it difficult for the audience to pinpoint the study findings. In addition, there is no link between the study analysis (i.e., section 5) and the discussion (i.e., section 6). It is unclear where the findings discussed in section 6 emerged from.

Our response: We have added one more section that prepares the reader for the Discussion and ties together the empirical chapter:

4.6 Two phases of social innovation implementation

Based on the examples of proactive CS at the factory and industry levels we have identified two phases of social innovation implementation: the ideation phase and testing/modification phase. In the ideation phase, represented by the factory level activities: Fair Living Wage Roadmap (see section 4.3) and Changing the suppliers’ mindsets (see section 4.4) the preparatory work is being carried out to lay the foundation for the testing/modification phase represented by industry level activities (see section 4.5). In the ideation phase, a broad behavioural change is taking place among multiple stakeholders involved while in the testing/modification phase the structural changes are prepared to gradually take place. We will discuss the critical factors pertinent to each phase in the next section.

  1. There is a need to provide clear justification as to how this study extends or contributes to sustainability or social innovation. 

Our response: We have added the justification:

“As discussed in the analysis section, we contribute to IB literature by theorizing about the role of social innovations as important means to drive change and push the CS agendas of MNE’s. While social innovations have been treated quite vaguely in extant literature, we outline a specific type of social innovation aimed at grand challenges and describe how this focus can be an important catalyst for proactive CS activities.”

  1. Abbreviations should be stated in full when first used within the manuscript — for example, H&M and EM. 

Our response: We have explained all abbreviations.

  1. Most references are dated. Consider using more recent references. 

Our response: We have added more recent sources which have helped us reinforce our theoretical arguments.

Chin, T., Yang, Y., Zhang, P., Yu, X., & Cao, L. (2019). Co-creation of social innovation: Corporate universities as innovative strategies for Chinese firms to engage with society. Sustainability, 11(5), 1438.

Nylund, P. A., Brem, A., & Agarwal, N. (2021). Innovation ecosystems for meeting sustainable development goals: The evolving roles of multinational enterprises. Journal of Cleaner Production, 281, 125329.

Prashantham, S., & Birkinshaw, J. (2020). MNE–SME cooperation: An integrative framework. Journal of International Business Studies, 51(7), 1161-1175.

Reviewer 4 Report

Dear Authors,

thank you for the opportunity to review the manuscript. I have read it with interest. The paper is well structured and the reasoning is clear. The topic is interesting, up to date and relevant. The overall impression of the paper is good. This allows to consider the paper suitable for publication.

However, I would recommend to consider the following minor changes, before publications:

  1. I would suggest to stress more added value of the paper in the abstract (the main contribution of the paper). The reader should be encouraged to read the paper.
  2. Method: Please provide at least one paragraph on research limitations.
  3. Some abbreviations used by the Authors are not introduced in the text (EM – line 75; HQ- line 284).
  4. Some editing errors were noted: lines195, 242, 409, 515, 574
  5. The references and citations are not adjusted to the Journal’s requirements.

Author Response

Proactive Corporate Sustainability via Social Innovation – a Case Study of Hennes & Mauritz’ Grand Challenge in Bangladesh

Letter to reviewers round 1

Dear reviewer!

Thank you for providing us with the valuable comments and suggestion on how to improve our paper. We have taken all your suggestions into consideration and hereafter address each reviewer separately.

Answer to reviewer 4

Thank you very much for positive and specific comments. We have addressed them in the following way:

  1. I would suggest to stress more added value of the paper in the abstract (the main contribution of the paper). The reader should be encouraged to read the paper.

Our response: We have changed the abstract in the way to add value to the paper and captures the key message more effectively:

“The purpose of the study is to explore how a multinational enterprise can use social innovations to drive change and solve grand challenges in an emerging market context. The paper brings market-shaping literature into a sustainability context, particularly by studying the implementation of social innovations in an emerging market context. Specifically, the study. involves an in-depth qualitative study of H&M’s fair living wages program in Bangladesh. We find that H&M is tackling utterances of grand challenges revealed by orchestrating social innovation in collaborations with local stakeholders. Social innovation is carried out in on-going projects involving multiple stakeholders. The study contributes to current literature by revealing that multinational enterprises indeed can use social innovation to drive change in emerging markets, although this requires long-term commitment, an ability and willingness to shape the surrounding business environment, and a prominent standing among key stakeholders”.

  1. Method: Please provide at least one paragraph on research limitations.

Our response: We have added a paragraph of research limitations in the method section, where we now say:

“We recognize that this single-case study of a firm in the fashion industry restricts us from generalizing results to other companies. For example, being a global consumer-oriented firm has certain implications on public scrutiny of CS operations, brand recognition, and social judgments in comparison with, for example, a business-to-business firm. Nonetheless, the quality and richness of data will allow us to capture a pertinent and emerging phenomenon and develop conceptualizations that could serve useful for future studies that seek to develop knowledge and theory about social innovation as a driver of CS development.  While consumer-oriented firms currently are seemingly taking the lead in this area, this is a growing phenomenon that over time could be applicable on a wider group of firms. “

3. Some abbreviations used by the Authors are not introduced in the text (EM – line 75; HQ- line 284).

Our response: We have explained all abbreviations.

4. Some editing errors were noted: lines195, 242, 409, 515, 574

Our response: We have checked the errors.

5. The references and citations are not adjusted to the Journal’s requirements.

Our response: We have started adjusting the references to the Journal’s format.

Reviewer 5 Report

The article pertains to the theme of the special issue titled "The Contribution of Innovation to CSR and Sustainability. Implementation by Firms in Emerging Markets". The content of the study closely corresponds to the title, and the problems discussed are presented in a clear and transparent manner, which makes them interesting for the reader. However, there are some minor adjustments to be taken into account, so I would consider a revision that substantively addresses the following points:

  • The summary should be revised as it does not contain clear definitions of: the main purpose, the methods, and procedures used, the main results and results, the conclusions from the data and results, the implications for further research.
  • The paper is quite poor in terms of the use of references to the literature dealing with the issues of social innovation – it is worth supplementing the theoretical framework with more references to this subject.
  • When presenting the results, the authors refer to the research question, but it is not clearly formulated in the text. What is the research question?
  • The research presented in the article concerns one company from the clothing industry, which does not allow for generalization of the insights resulting from the Author(s) observations. The article should be supplemented with clear explanations of how the results can be compared with other MNEs.
  • The drawings included in the text are quite extensive. I suggest to reduce them and standardise the fonts.
  • Finally, I recommend that the authors mention the limitations and possible direction of development of their research.

Author Response

Proactive Corporate Sustainability via Social Innovation – a Case Study of Hennes & Mauritz’ Grand Challenge in Bangladesh

Letter to reviewers round 1

Dear reviewer!

Thank you for providing us with the valuable comments and suggestion on how to improve our paper. We have taken all your suggestions into consideration and hereafter address each reviewer separately.

Answer to reviewer 5

Thank you very much for positive and specific comments. We have addressed them in the following way:

  1. The summary should be revised as it does not contain clear definitions of: the main purpose, the methods, and procedures used, the main results and results, the conclusions from the data and results, the implications for further research.

Our response: We have changed the abstract to consider all the points you have mentioned. The abstract now says:

“The purpose of the study is to explore how a multinational enterprise can use social innovations to drive change and solve grand challenges in an emerging market context. The paper brings market-shaping literature into a sustainability context, particularly by studying the implementation of social innovations in an emerging market context. Specifically, the study. involves an in-depth qualitative study of H&M’s fair living wages program in Bangladesh. We find that H&M is tackling utterances of grand challenges revealed by orchestrating social innovation in collaborations with local stakeholders. Social innovation is carried out in on-going projects involving multiple stakeholders. The study contributes to current literature by revealing that multinational enterprises indeed can use social innovation to drive change in emerging markets, although this requires long-term commitment, an ability and willingness to shape the surrounding business environment, and a prominent standing among key stakeholders”.

2. The paper is quite poor in terms of the use of references to the literature dealing with the issues of social innovation – it is worth supplementing the theoretical framework with more references to this subject.

Our response: We have added more sources on social innovation, see section 2.3. where you now find this text:

“Recently, literature has drawn a line between social innovation and efforts to proactively tackle grand challenges, for example achieving UN’s social development goals (Nylund et al., 2021). Prashantham and Birkinshaw (2020) also argue that MNEs can benefit societies - and themselves - by cooperating with local companies in EMs where addressing SDGs is a pressing concern. Developing local relationships for this purpose would allow firms to combine complementary resources to address societal and economic needs simultaneously.  Correspondingly, Nylund et al. (2021) argue that to develop social innovations, MNEs form business ecosystems and networks involving local stakeholders that influence not only business but wider social systems. However, literature has not yet fully accounted for this diversity of social innovation, specifically related to the different types of social innovations as well as how social innovations can serve (or not serve) proactive efforts of CS”.

  1. When presenting the results, the authors refer to the research question, but it is not clearly formulated in the text. What is the research question?

Our response: We have clarified the purpose of the study used instead of research question:

“The purpose of this study is to explore how a MNE can use social innovations to drive change and solve grand challenges in an EM context”

4. The research presented in the article concerns one company from the clothing industry, which does not allow for generalization of the insights resulting from the Author(s) observations. The article should be supplemented with clear explanations of how the results can be compared with other MNEs.

Our response: We have discussed the limitations of the study, see this new text added to the method section:

“We recognize that this single-case study of a firm in the fashion industry restricts us from generalizing results to other companies. For example, being a global consumer-oriented firm has certain implications on public scrutiny of CS operations, brand recognition, and social judgments in comparison with, for example, a business-to-business firm. Nonetheless, the quality and richness of data will allow us to capture a pertinent and emerging phenomenon and develop conceptualizations that could serve useful for future studies that seek to develop knowledge and theory about social innovation as a driver of CS development.  While consumer-oriented firms currently are seemingly taking the lead in this area, this is a growing phenomenon that over time could be applicable on a wider group of firms”.

5. The drawings included in the text are quite extensive. I suggest to reduce them and standardise the fonts.

Our response: We agree that the drawings are extensive. We have standardised the fonts but still left the findings to make the big picture more visual.

6. Finally, I recommend that the authors mention the limitations and possible direction of development of their research.

Our response: We have added the limitations and possible directions of research in the “Conclusion” section of the manuscript.

“This study represents an early attempt to delve into the underpinnings of the actual implementation of mission-oriented innovation in EM settings by one particular MNE and this way is limited in terms of generalizability to other MNEs PS implementation in EM. However, we suggest that some key analytical insights might be relevant for MNEs in general. For example, we find that social processes (the ideation phase and the testing/modification phase) are instrumental in implementing these social innovation regimes.”

And, in this section we now also highlight the following:

“Based on this study, we call for further research and conceptualizations that consider the modes of governance around social innovations, e.g., ecosystems that emerge around social innovations in various settings. A topic related to social innovation in ecosystems worth further scholarly attention is the notion of orchestration. How can MNEs orchestrate change in ecosystem settings? What external circumstances and internal organizational assets are favourable for orchestration? We also encourage researchers to delve deeper into the issue of market conflict that could surface because of launching social innovations in market settings where MNEs’ sustainability principles clash with local norms (which we have seen some evidence of in this study).”

Round 2

Reviewer 2 Report

The authors have addressed all my comments for this manuscript.